# The IgA in milk induced by SARS-CoV-2 infection is comprised of mainly secretory antibody that is neutralizing and highly durable over time

Alisa Fox[1], Jessica Marino[2,3], Fatima Amanat[4,5], Kasopefoluwa Y. Oguntuyo[4,5], Jennifer Hahn-Holbrook[2,3], Benhur Lee[4], Susan Zolla-Pazner[1], Rebecca L. Powell[1]*

1 Division of Infectious Diseases, Department of Medicine, Icahn School of Medicine at Mount Sinai, New York, New York, United States of America, 2 Health Sciences Research Institute, University of California Merced, Merced, California, United States of America, 3 Department of Psychology, University of California Merced, Merced, California, United States of America, 4 Department of Microbiology, Icahn School of Medicine at Mount Sinai, New York, New York, United States of America, 5 Graduate School of Biomedical Sciences, Icahn School of Medicine at Mount Sinai, New York, New York, United States of America

* Rebecca.Powell@mssm.edu

**Data Availability Statement:** All relevant data are within the paper and its Supporting information files.

## Abstract

Approximately 10% of infants infected with SARS-CoV-2 will experience COVID-19 illness requiring advanced care. A potential mechanism to protect this population is passive immunization via the milk of a previously infected person. We and others have reported on the presence of SARS-CoV-2-specific antibodies in human milk. We now report the prevalence of SARS-CoV-2 IgA in the milk of 74 COVID-19-recovered participants, and find that 89% of samples are positive for Spike-specific IgA. In a subset of these samples, 95% exhibited robust IgA activity as determined by endpoint binding titer, with 50% considered high-titer. These IgA-positive samples were also positive for Spike-specific secretory antibody. Levels of IgA antibodies and secretory antibodies were shown to be strongly positively correlated. The secretory IgA response was dominant among the milk samples tested compared to the IgG response, which was present in 75% of samples and found to be of high-titer in only 13% of cases. Our IgA durability analysis using 28 paired samples, obtained 4–6 weeks and 4–10 months after infection, found that all samples exhibited persistently significant Spike-specific IgA, with 43% of donors exhibiting increasing IgA titers over time. Finally, COVID-19 and pre-pandemic control milk samples were tested for the presence of neutralizing antibodies; 6 of 8 COVID-19 samples exhibited neutralization of Spike-pseudotyped VSV (IC$_{50}$ range, 2.39–89.4ug/mL) compared to 1 of 8 controls. IgA binding and neutralization capacities were found to be strongly positively correlated. These data are highly relevant to public health, not only in terms of the protective capacity of these antibodies for breastfed infants, but also for the potential use of such antibodies as a COVID-19 therapeutic, given that secretory IgA is highly in all mucosal compartments.

**Funding:** RP is supported by the NIH/NIAID grant number R01AI158214-01 The funders had no role in study design, data collection and analysis, decision to publish, or preparation of the manuscript.

**Competing interests:** The authors have declared that no competing interests exist.

# Background

Though COVID-19 pathology among children is typically more mild compared to adults, approximately 10% of infants under the age of one year experience severe COVID-19 illness requiring advanced care, and an ever-growing number of children appear to exhibit signs of "Multisystem Inflammatory Syndrome in Children (MIS-C) associated with COVID-19" weeks or months after exposure [1, 2]. Furthermore, infants and young children can also transmit SARS-CoV-2 to others and the efficacy of vaccines available for adults have not yet been evaluated for young children or infants [3]. Certainly, protecting this population from infection is essential [4].

One potential mechanism of protection is passive immunity provided through breastfeeding by a previously infected mother. Mature human milk contains ~0.6mg/mL of total immunoglobulin [5]. Approximately 90% of human milk antibody (Ab) is IgA, nearly all in secretory (s) form (sIgA, which consists of polymeric Abs complexed to J-chain and secretory component (SC) proteins) [6]. Nearly all sIgA derives from the gut-associated lymphoid tissue (GALT), via the *entero-mammary link*, though there is also homing of B cells from other mucosa (e.g., from the respiratory system), and possibly drainage from local lymphatics of systemic IgA to the mammary gland [6]. Unlike the Abs found in serum, sIgA in milk is highly stable and resistant to enzymatic degradation not only in milk and the infant mouth and gut, but in all mucosae including the gastrointestinal tract, upper airway, and lungs [7]. Notably, it has been shown that after 2 hours in the infant stomach, the total IgA concentration decreases by <50%, while IgG concentration decreases by >75% [8].

Previously we reported on 15 milk samples obtained early in the pandemic from donors recently-recovered from a confirmed or suspected case of COVID-19 [9]. In that preliminary study, it was found that all samples exhibited significant IgA binding activity against the SARS-CoV-2 Spike. Eighty percent of samples further tested for Ab binding reactivity to the receptor binding domain (RBD) of the Spike exhibited significant IgA binding, and all of these samples were also positive for RBD-specific secretory Ab reactivity with only small subsets of samples exhibiting specific IgG and/or IgM activity, strongly suggesting the RBD-specific IgA was sIgA. In the present study, we report on the prevalence and isotypes of Spike-specific milk Ab from a larger cohort of donors obtained 4–6 weeks post-confirmed SARS-CoV-2 infection, on the durability of these Abs up to 10 months post-infection, and on SARS-CoV-2-directed neutralization by Abs in a subset of these samples.

# Methods

## Study participants

This study was approved by the Institutional Review Board (IRB) at Mount Sinai Hospital (IRB 19–01243). Individuals were eligible to have their milk samples included in this analysis if they were lactating and had a confirmed SARS-CoV-2 infection (by an FDA-approved COVID-19 PCR test) 4–6 weeks prior to the initial milk sample used for analysis. This post-infection window was selected so as to minimize any contact with participants or their samples when they might have been contagious to the research team, while still capturing the reported peak period for SARS-CoV-2 Ab responses [10]. Participants were excluded if they had any acute or chronic health conditions affecting the immune system. Participants were recruited nationally via social media in April-June of 2020 and subject to an informed consent process. Certain participants contributed milk they had previously frozen for personal reasons, while most pumped samples specifically for this research project. All participants were either asymptomatic or experienced mild-moderate symptoms of COVID-19 that were managed at home. Participants were asked

to collect approximately 30mL of milk per sample into a clean container using electronic or manual pumps, and if able and willing, to continue to pump and save monthly milk samples after the initial sample as part of our longitudinal analysis. If any of the participants submitted longitudinal samples at least 4 months after their initial sample, those samples were also included in the present analysis. As little longitudinal mucosal Ab data in COVID-19-recovered individuals past 3 months has been reported to date, the ≥4 month time point was selected, and as many samples that were available were used. To estimate the proportion (p) of all COVID-19-recovered milk donors that would exhibit positive IgA titers against SARS-CoV-2 in their milk after infection, we determined based on the reported IgG seroconversion rate of 90% after mild SARS-CoV-2 infection [11], that the precision (d) of the 95% confidence interval (CI) for p (CI = [p+-d]), as a function of the cohort size N of 74 would allow us to estimate p with 6.79% error. In terms of the cohort size N of 20 for milk IgG and secretory Ab analyses, this would allow us to estimate p with 13.15% error.

Milk was frozen in participants' home freezers until samples were picked up and stored at -80°C until Ab testing. Pre-pandemic negative control milk samples were obtained in accordance with IRB-approved protocol 17–01089 prior to December 2019 from healthy lactating women in New York City, and had been stored in laboratory freezers at -80°C before processing following the same protocol described for COVID-19 milk samples. All demographic information on participant milk samples is shown in Table 1. Given the diversity of participant ages and stages of lactation, this study sample can be considered representative of a larger population. Notably, 67% of COVID-19-recovered participants reported their race/ethnicity as white or Caucasian, and therefore this sample set is not diverse enough to be considered representative of the USA as a whole. More work needs to be done to obtain sufficient samples from non-white participants. Ten COVID-19-recovered (COV101-COV117) and 10 pre-pandemic control (NEG046-NEG059) participants included in the present study also had their Spike IgA ELISA data reported in the our pilot study publication [9].

## ELISA

Levels of SARS-CoV-2 Abs in human milk were measured as previously described [9]. Briefly, before Ab testing, milk samples were thawed, centrifuged at 800g for 15 min at room temperature, fat was removed, and the de-fatted milk transferred to a new tube. Centrifugation was repeated 2x to ensure removal of all cells and fat. Skimmed acellular milk was aliquoted and frozen at -80°C until testing. Both COVID-19 recovered and control milk samples were then tested in separate assays measuring IgA, IgG, and secretory-type Abs, in which the secondary Ab used for the latter measurement was specific for free and bound SC. Half-area 96-well plates (Fisher cat# 14-245-153) were coated with the full trimeric recombinant Spike protein produced as described previously [12]. Plates were incubated at 4°C overnight, washed in 0.1% Tween 20/PBS (PBS-T), and blocked in PBS-T/3% goat serum (Fisher cat# PCN5000)/0.5% milk powder (Fisher cat# 50-751-7665) for 1 h at room temperature. Milk was used undiluted or titrated 4-fold in 1% bovine serum albumin (BSA; Fisher cat# 50-105-8877)/PBS and added to the plate. After 2h incubation at room temperature, plates were washed and incubated for 1h at room temperature with horseradish peroxidase-conjugated goat anti-human-IgA, goat anti-human-IgG (Fisher cat# 40-113-5 and #OB201405), or goat anti-human-secretory component (MuBio cat# GAHu/SC/PO) diluted in 1% BSA/PBS. Plates were developed with 3,3',5,5'-Tetramethylbenzidine (TMB; Fisher cat#PI34028) reagent followed by 2N sufuric acid (Fisher cat# MSX12446) and read at 450nm on a BioTek Powerwave HT plate reader. Assays were performed in duplicate and repeated 2x.

**Table 1. Participant data.**

| Sample ID | Age | Race/ethnicity | State of residence | Months Postpartum (1st sample) |
|---|---|---|---|---|
| COV101 | 32 | White or Caucasian | New York | 4 |
| COV102 | 30 | NR | New York | <1* |
| COV103c | 25 | Hispanic or Latino | New York | 4 |
| COV108b | 26 | Black or African American | New York | 5 |
| COV109b | 32 | Asian or Pacific Islander | New York | 3 |
| COV110 | 32 | White or Caucasian | New York | 2 |
| COV112 | 33 | White or Caucasian | New York | <1* |
| COV113 | 27 | Hispanic or Latino | New York | 7 |
| COV116 | 34 | White or Caucasian | New York | 7 |
| COV117 | 32 | White or Caucasian | New York | 3 |
| COV119 | 31 | White or Caucasian | New York | 1* |
| COV120 | 32 | White or Caucasian | New York | <1* |
| COV121 | 39 | Asian or Pacific Islander | New York | 23 |
| COV122b | 36 | White or Caucasian | New York | <1* |
| COV123b | 33 | White or Caucasian | New Jersey | 9 |
| COV124 | 44 | Multiracial or biracial | New York | 16 |
| COV125b | 21 | White or Caucasian | New York | 6 |
| COV126b | 26 | Hispanic or Latino | New York | 6 |
| COV127 | 34 | Asian or Pacific Islander | New York | 8 |
| COV128b | 22 | Multiracial or biracial | New York | 7 |
| COV129c | 31 | Asian or Pacific Islander | New York | 2 |
| COV130b | 24 | NR | New York | 13 |
| COV131a | 21 | White or Caucasian | New York | 3 |
| COV132a | 32 | White or Caucasian | Tennessee | 10 |
| COV133a | 20 | Black or African American | New York | <1* |
| COV134a | 26 | White or Caucasian | New York | <1* |
| COV135a | 35 | White or Caucasian | New York | 6 |
| COV136a | 36 | White or Caucasian | New York | <1* |
| COV137a | 32 | White or Caucasian | New York | |
| COV142a | 35 | White or Caucasian | New York | 14 |
| COV143a | 37 | White or Caucasian | Connecticut | 8 |
| COV144b | 31 | White or Caucasian | New Jersey | 5 |
| COV146b | 30 | White or Caucasian | New York | 2 |
| COV147b | 25 | White or Caucasian | New York | 2 |
| COV148a | 37 | Black or African American | New York | 5 |
| COV150a | 32 | White or Caucasian | New York | 4 |
| COV153a | 35 | White or Caucasian | North Carolina | 10 |
| COV154a | 36 | White or Caucasian | New York | 2 |
| COV155a | 40 | White or Caucasian | New York | <1* |
| COV159a | 39 | NR | New York | 11 |
| COV162c | 28 | White or Caucasian | New York | 5 |
| COV163a | 32 | NR | New York | 2 |
| COV165a | 33 | White or Caucasian | Ohio | 8 |
| COV167a | 29 | Asian or Pacific Islander | New York | 24 |
| COV168d | 23 | White or Caucasian | New York | 4 |
| COV169a | 33 | White or Caucasian | New York | 10 |
| COV171b | 34 | Hispanic or Latino | New York | 8 |
| COV172b | 22 | White or Caucasian | New York | 1 |

*(Continued)*

**Table 1.** (Continued)

| Sample ID | Age | Race/ethnicity | State of residence | Months Postpartum (1st sample) |
|---|---|---|---|---|
| COV175a | 32 | White or Caucasian | New York | 2 |
| COV176a | 29 | White or Caucasian | New Jersey | <1* |
| COV177c | 37 | White or Caucasian | New York | 8 |
| COV181a | 29 | Asian or Pacific Islander | New York | <1* |
| COV183a | 38 | White or Caucasian | New York | 2 |
| COV184a | 37 | Asian or Pacific Islander | New York | 3 |
| COV185d | 34 | White or Caucasian | New York | 12 |
| COV186a | 32 | White or Caucasian | New Jersey | 19 |
| COV187a | 35 | White or Caucasian | New York | 8 |
| COV188a | 36 | White or Caucasian | New York | 4 |
| COV189a | 36 | Asian or Pacific Islander | California | 3 |
| COV190a | 34 | White or Caucasian | New York | 2 |
| COV192a | 35 | White or Caucasian | New York | <1* |
| COV204a | 36 | White or Caucasian | Maryland | 8 |
| COV207a | 27 | White or Caucasian | New York | 16 |
| COV208a | 34 | White or Caucasian | Washington DC | 6 |
| COV220a | 31 | White or Caucasian | New York | 1* |
| COV221a | 31 | White or Caucasian | New York | 3 |
| COV222a | 33 | NR | New York | <1* |
| COV223a | 40 | Multiracial or biracial | New York | 3 |
| COV224a | 32 | Asian or Pacific Islander | New York | 5 |
| COV225a | 36 | White or Caucasian | New York | <1* |
| COV226a | 38 | White or Caucasian | New York | 11 |
| COV227a | 28 | White or Caucasian | New York | 6 |
| COV228a | 34 | White or Caucasian | New York | 6 |
| COV229a | 33 | White or Caucasian | New York | 2 |
| NEGS032 | 38 | Asian or Pacific Islander | New York | 3 |
| NEGS034 | 36 | Hispanic or Latino | New York | 3 |
| NEGS036 | 32 | White or Caucasian | New York | 5 |
| NEGS046 | 38 | Hispanic or Latino | New York | 7 |
| NEGS048 | 31 | White or Caucasian | New York | 10 |
| NEGS050 | 38 | Asian or Pacific Islander | New York | 9 |
| NEGS051 | 25 | White or Caucasian | New York | 9 |
| NEGS052 | 40 | White or Caucasian | New York | 7 |
| NEGS054 | 28 | White or Caucasian | New York | 6 |
| NEGS055 | 27 | Asian or Pacific Islander | New York | 7 |
| NEGS056 | 39 | Hispanic or Latino | New York | 8 |
| NEGS058 | 39 | Hispanic or Latino | New York | 5 |
| NEGS059 | 38 | White or Caucasian | New York | 7 |
| NEGS081 | 28 | White or Caucasian | New York | 3 |
| NEGS086 | 34 | White or Caucasian | New York | 14 |
| NEGS088 | 27 | White or Caucasian | New York | 5 |
| NEGS090 | 34 | White or Caucasian | New York | 1 |
| NEGS092 | 23 | Black or African American | New York | 8 |
| NEGS093 | 39 | White or Caucasian | New York | 4 |
| NEGS094 | 33 | White or Caucasian | New York | 6 |

*infected antepartum;

NR: not reported.

## IgA extraction from milk

Total IgA was extracted from 25—100mL of milk using peptide M agarose beads (Fisher cat# NC0127215) following manufacturer's protocol, concentrated using Amicon Ultra centrifugal filters (10 kDa cutoff; Fisher cat# UFC901008) and quantified by Nanodrop.

## Pseudovirus neutralization assay

Neutralization assays were performed using a standardized SARS-CoV-2 Spike-pseudotyped Vesicular Stomatitis Virus (VSV)-based assay with ACE2- and TMPRSS2-expressing 293T cells (clone F8-2; ATCC CRL-3216-derived) as previously described [13]. This cell line was routinely verified for consistent ACE2 and TMPRSS2 expression by flow cytometry as well as by inclusion of assay-to-assay control virus to monitor consistent infection levels. Pseudovirus was produced by transfection of 293T cells with SARS-CoV-2 Spike plasmid, followed 8 h later by infection with a VSVΔG-rLuc reporter virus. Two days post-infection, supernatants were collected and clarified by centrifugation [13]. Cells and viruses were prepared by and obtained from the Benhur Lee lab. A consistent,pre-titrated amount of pseudovirus was incubated with serial dilutions of extracted IgA for 30 min at room temperature prior to infection of cells seeded the previous day. Twenty hours post-infection, cells were processed and assessed for luciferase activity as described [13].

## Analytical methods

Control milk samples obtained prior to December 2019 were used to establish positive cutoff values for each assay. Milk was defined as positive for the SARS-CoV-2 Abs if optical density (OD) values measured using undiluted milk from COVID-19-recovered donors were two standard deviations (SD) above the mean ODs obtained from control samples. Endpoint dilution titers were determined from log-transformed titration curves using 4-parameter non-linear regression and an OD cutoff value of 1.0. Endpoint dilution positive cutoff values were determined as above. Percent neutralization was calculated as (1- (average luciferase Relative Light Units (RLU) of triplicate test wells–average luciferase expression RLU of 6 'virus only' control wells) *100. Mann-Whitney U tests were used to assess significant differences between unpaired grouped data. Paired Student's t-test was used to assess significant differences between longitudinal time points. The concentration of milk IgA required to achieve 50% neutralization ($IC_{50}$) was determined as described above for endpoint determination. Correlation analyses were performed using Spearman correlations. All statistical tests were performed in GraphPad Prism, were 2-tailed, and significance level was set at p-values $< 0.05$.

## Results

### Ab profile in milk from COVID-19-recovered donors 4–6 weeks after infection

Sixty-six of 74 samples (89%) were positive for Spike-specific IgA, with the COVID-19 samples exhibiting significantly higher Spike-specific IgA binding compared to controls (Fig 1a; p<0.0001). Following this initial screening, 40 of the Spike-positive samples were further titrated to determine binding endpoint titers as an assessment of Ab affinity and/or quantity (Fig 1b). Thirty-eight of 40 (95%) Spike-reactive samples exhibited positive IgA endpoint titers and 19 of these samples (50%) were ≥5 times higher than the endpoint titer of the positive cutoff value, and were therefore designated as 'high-titer' (Fig 1c).

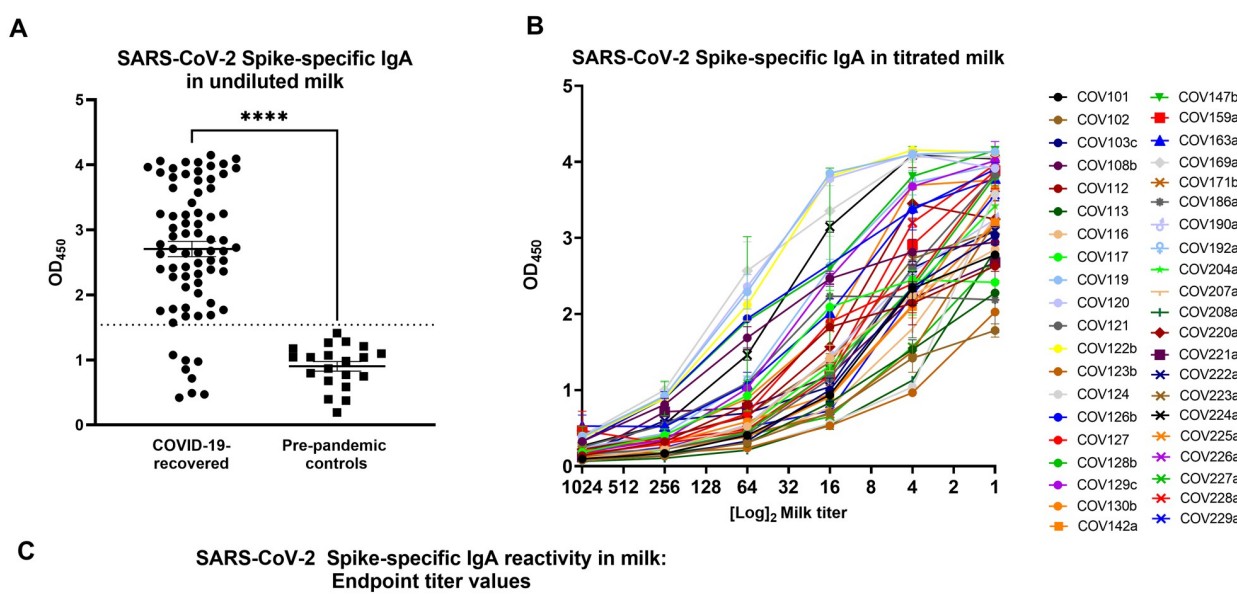

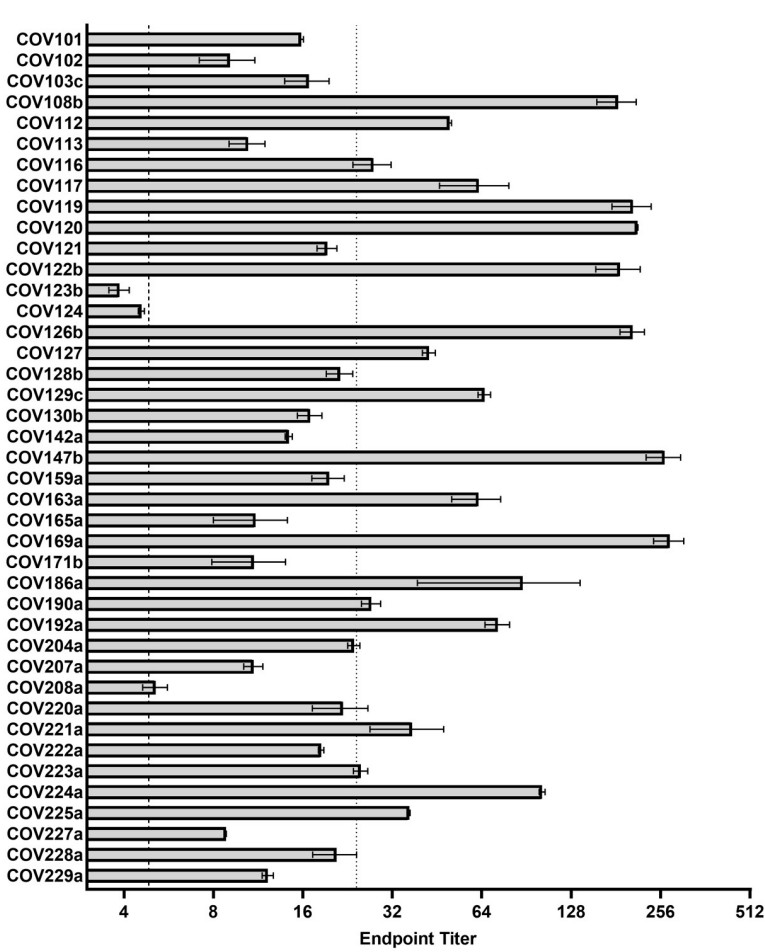

**Fig 1. A robust, Spike-specific IgA response in milk commonly occurs after SARS-CoV-2 infection.** (A) Screening of undiluted milk samples for specific IgA by ELISA against the full-length Spike trimer. COVID-19 group, N = 74; control group N = 20. Mean values with SEM are shown. Dotted line: positive cutoff value (mean OD of negative control milk samples + 2*SD). ****p<0.0001. Mann-Whitney U test (2-tailed) was used to compare grouped data with significance level set at p < 0.05. (B) Full titration against Spike of 40 milk samples found to be positive by the initial screening. (C) Endpoint dilution titers of the 40 titrated milk samples. Segmented line: positive cutoff value; dotted line: 5x positive endpoint cutoff value, designating samples as 'high-titer'. Mean values with SEM are shown.

Additionally, 20 samples assayed for Spike-specific IgA were also assessed for Spike-specific secretory Ab (by detecting for SC), and IgG. Nineteen of these undiluted milk specimens (95%) from convalescent COVID-19 donors were positive for Spike-specific secretory Abs compared to pre-pandemic control milk (Fig 2a). One sample (COV125b) was negative for

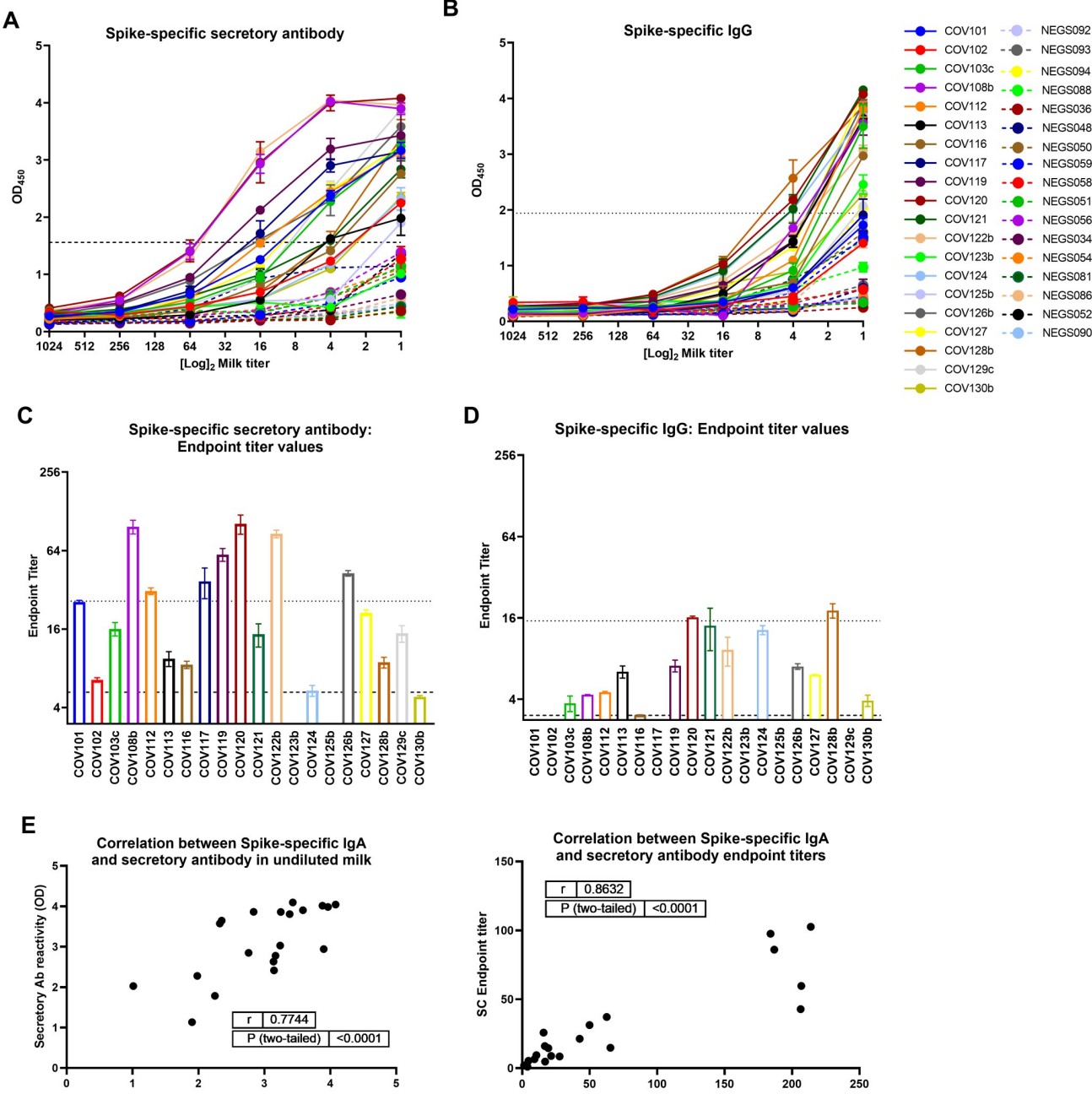

**Fig 2. The dominant Spike-specific IgA response in milk after SARS-CoV-2 infection is strongly correlated with a robust secretory Ab response, while specific IgG activity is relatively modest.** Twenty samples assayed for Spike-specific IgA were also assessed for Spike-specific secretory Ab (by detecting for SC), and IgG. (A, B) Full titration against Spike, detecting (A) secretory Ab, and (B) IgG. NEG (i.e. negative)/segmented lines: pre-pandemic controls. COV/solid lines: milk from COVID-19-recovered donors. Dotted lines: positive cutoff values. (C, D) Endpoint titer values calculated for (A) secretory Ab, and (B) IgG. Segmented lines: positive cutoff values; dotted lines: 5x positive cutoff (high-titer cutoff). (E) IgA and secretory Ab binding OD values or endpoint titers were used in 2-tailed Spearman correlation tests. SC: secretory component.

specific IgA but positive for specific secretory Ab, while another sample (COV123b) was positive for specific IgA but negative for specific secretory Ab. Eighteen undiluted milk samples (95%) exhibiting Spike-specific secretory Ab activity also exhibited positive endpoint titers (Fig 2c). Of the samples found to be high-titer for Spike-specific IgA, 7 were also high-titer for specific secretory Ab (70%). Mean OD values for undiluted milk and endpoint titers were used in separate Spearman correlation tests to compare IgA and secretory Ab reactivity (Fig 2e). It was found that IgA and secretory Ab levels were positively correlated (using ODs: r = 0.77, p<0.0001; using endpoint titers: r = 0.86, p<0.0001). Additionally, 15/20 undiluted milk samples from COVID-19-recovered donors were positive for Spike-specific IgG compared to pre-pandemic controls (75%; Fig 2b), with 13/15 of these samples exhibiting a positive endpoint titer (87%; Fig 2d), and 2/15 designated as high titer with values ≥5 times cutoff (13%). No correlation was found between IgG and IgA titers or between IgG and SC titers (S1 Fig).

## Durability of the SARS-CoV-2 Spike-specific milk IgA response

To assess the durability of this sIgA-dominant response, 28 pairs of milk samples obtained from COVID-19-recovered donors 4–6 weeks and 4–10 months after infection were assessed for Spike-specific IgA. All donors exhibited persistently significant Spike-specific IgA titers at the follow-up time point. Mean endpoint titers from the early to the late milk samples grouped were not significantly different (Fig 3a). Fourteen donors (50%) exhibited >10% decrease in IgA titer, 12 donors (43%) exhibited >10% increase in IgA titer, and 2 donors (7%) exhibited no change in titer (Fig 3a). Notably, only 2 donors (7%) exhibited >50% decrease in titer over time. Furthermore, examining a subset of 14 of these samples with the longest follow-up, obtained 7–10 months after infection, mean endpoint titers measured from the early to the late milk samples were also not significantly different (19.8 and 17.8, respectively; Fig 3b). These longest follow-up samples included 4 donors (29%) with >10% decrease in IgA titer, 7 donors (50%) with >10% increase in IgA titer, and 3 donors (21%) with no change in titer (Fig

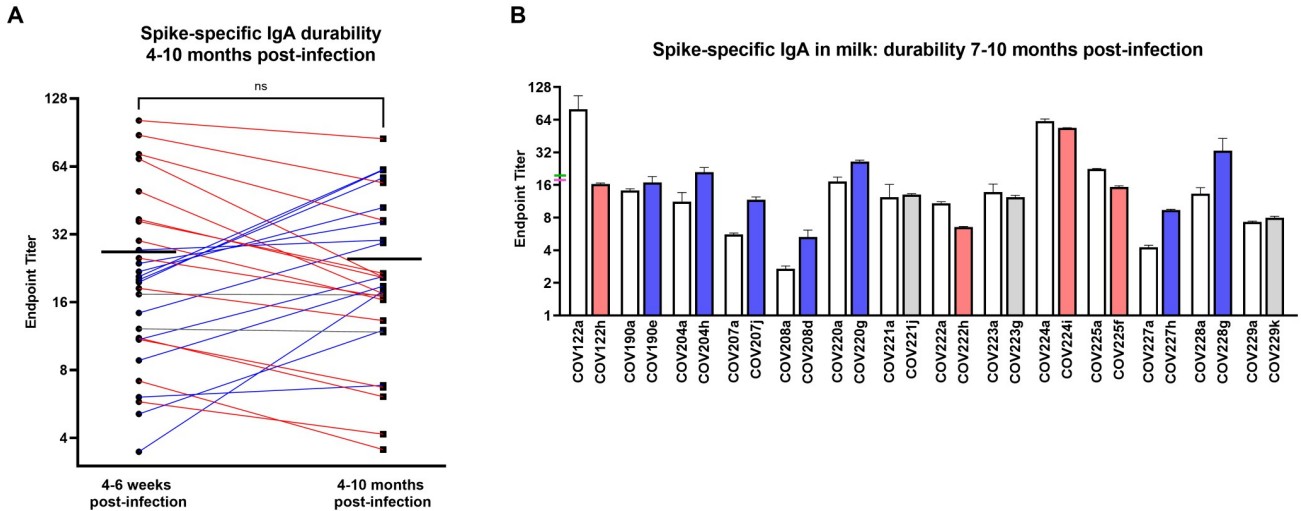

**Fig 3. The Spike-specific IgA response in milk after SARS-CoV-2 infection is highly durable over time.** (A) IgA endpoint titers determined from Spike ELISA for 28 pairs of milk samples obtained from COVID-19-recovered donors 4–6 weeks and 4–10 months after infection are shown. Mean endpoint values for each group are shown. Blue lines: >10% increase, red lines: >10% decrease, grey lines: <10% change. NS: not significant. A paired t-test (2-tailed) was used to assess significance. (B) IgA endpoint titers for a subset of 14 paired samples obtained 4–6 weeks and 7–10 months after infection. Mean with SEM is shown. Mean endpoint values for the 4–6 week and 7–10 month groups are indicated on the y-axis as green and pink ticks, respectively. Blue bars: >10% increase, red bars: >10% decrease, grey bars: <10% change.

3b). Only 1 donor (7%) exhibited >50% decrease in titer, and all donors exhibited persistently significant Spike-specific IgA titers.

## SARS-CoV-2 neutralization capacity of total milk IgA from COVID-19-recovered donors

Total IgA was extracted from 8 COVID-19 samples obtained 4–6 weeks after infection and 8 control milk samples previously analyzed for their Spike-specific Ab profile. All 8 COVID-19 samples had been shown to exhibit positive Spike-specific IgA and secretory Ab titers (Figs 1 and 2). Neutralization capacity was tested using a Vesicular Stomatitis Virus (VSV)-based pseudovirus assay, wherein the native VSV surface protein G is replaced by the SARS-CoV-2 Spike, as described previously ((13); Fig 4). At the maximum concentration tested (200ug/mL total

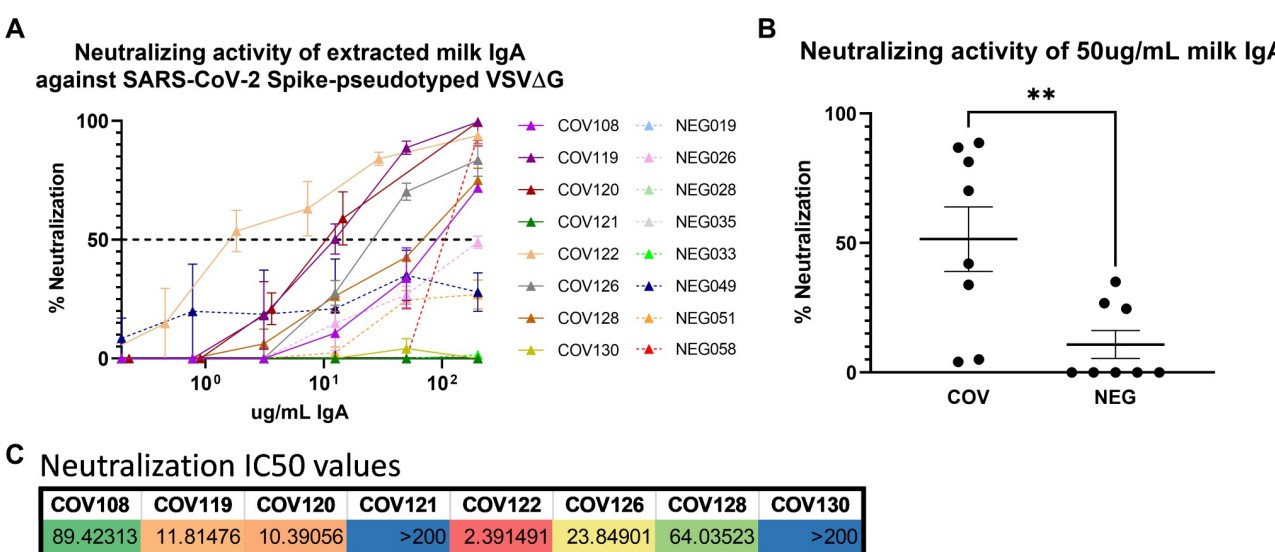

### C Neutralization IC50 values

| COV108 | COV119 | COV120 | COV121 | COV122 | COV126 | COV128 | COV130 |
|---|---|---|---|---|---|---|---|
| 89.42313 | 11.81476 | 10.39056 | >200 | 2.391491 | 23.84901 | 64.03523 | >200 |
| **NEG019** | **NEG026** | **NEG028** | **NEG033** | **NEG035** | **NEG038** | **NEG049** | **NEG051** |
| >200 | >200 | >200 | >200 | >200 | 178.6628 | >200 | >200 |

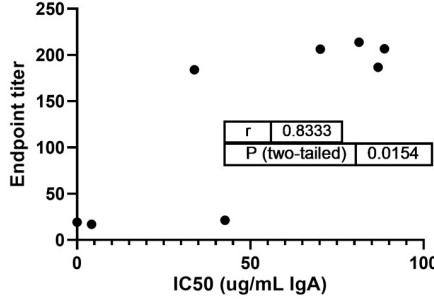

**Fig 4. Extracted milk IgA from COVID-19-recovered donors exhibits SARS-CoV-2 Spike-targeted neutralization potency that is highly correlated with IgA binding activity.** (A) Total IgA was purified from 8 COVID-19 and 8 control milk samples by conventional means using peptide M agarose. IgA was titrated and tested in a VSV-based SARS-CoV-2 pseudovirus neutralization assay. NEG/segmented lines: pre-pandemic controls. COV/solid lines: COVID-19-recovered milk samples. Segmented line: 50% neutralization cutoff value. (B) Percent neutralization achieved using 50ug/mL of total extracted milk IgA. Mean values with SEM are shown. **p = 0.0064. Mann-Whitney U test (2-tailed) was used to compare grouped data with significance level set at p < 0.05. (C) Neutralization IC50 values determined from IgA titration curves. (D) Endpoint titer values determined in Fig 1 and IC50 values were used in a 2-tailed Spearman correlation test.

purified milk IgA), 6/8 (75%) COVID-19 samples exhibited >50% neutralization (mean, 87% neutralization; range, 70%–100%), while only 1/8 control samples (13%) achieved this benchmark (94% neutralization; Fig 4a). Mean percent neutralization values at 50ug/ml extracted IgA were grouped and compared among COVID-19 and pre-pandemic control samples. COVID-19 samples exhibited significantly greater neutralization compared to controls (p = 0.0064; Fig 4b). As well, when the concentration of IgA required to achieve 50% neutralization (IC50) was determined, 7/8 pre-pandemic controls did not achieve 50% neutralization (IC50>200ug/mL while, for the COVID-19 samples, 2/8 did not achieve 50% neutralization, and the mean IC50 for the 6 COVID19 specimens that displayed neutralizing activity was 33.6ug/mL of total IgA (range, 2.39–89.4ug/mL; Fig 4c). Finally, we compared the neutralization IC50 titers to the IgA endpoint titers measured for these samples (Fig 1). There was a significant positive correlation between IgA binding and neutralization capacities (r = 0.83, p = 0.0154; Fig 4d). Notably, the 2 non-neutralizing COVID-19 IgA samples also exhibited the lowest IgA endpoint titers (COV121, COV130; mean IgA endpoint titers of 19 and 17, respectively), while the 6 neutralizing samples exhibited high Spike-specific IgA binding titers (Figs 1c and 4c).

## Discussion

There has been no evidence that SARS-CoV-2 transmits via human milk, with sporadic cases of viral RNA (not infectious particles) detected on breast skin [14]; however, there have been reports of viral RNA in the milk (reviewed in [15]), though collection methods in these reports did not necessarily include masking, cleaning of the breast, or even handwashing to avoid contamination from the donor's environment. As such, the WHO and CDC recommend that infants not be separated from SARS-CoV-2-infected mothers, and that breastfeeding should be established and not disrupted (depending on the mothers' desire to do so), in combination with masking and other hygiene efforts [16, 17].

We and others have reported SARS-CoV-2-specific Abs in milk obtained from donors with previously confirmed or suspected infection [9, 14, 18, 19]. Here, we have significantly expanded our earlier work in which we reported on SARS-CoV-2 Ab prevalence among 75 COVID-19-recovered participants whose milk samples were obtained 4–6 weeks after confirmed SARS-CoV-2 infection. Indeed, we have confirmed among this much larger sample set that a SARS-CoV-2 IgA Ab response in milk after infection is very common. Our analysis of a subset of 20 milk samples from COVID-19-recovered participants suggests that this IgA response dominates compared to the measurable but relatively lower titer IgG response. Importantly, a very strong positive correlation was found between Spike-specific milk IgA and secretory Abs, using both ELISA OD values of Ab binding in undiluted milk as well as Ab binding endpoint titers, indicating that a very high proportion of the SARS-CoV-2 Spike-specific IgA measured in milk after SARS-CoV-2 infection is sIgA, confirming our early reports. This is relevant for the effective protection of a breastfeeding infant, given the high durability of secretory Abs in the relatively harsh mucosal environments of the infant mouth and gut [7, 8]. These data are also relevant to the possibility of using extracted milk IgA as a COVID-19 therapy. Extracted milk sIgA used therapeutically would likely survive well upon targeted respiratory administration, with a much lower dose of Ab likely needed for efficacy compared to systemically-administered convalescent plasma or purified plasma immunoglobulin.

All COVID-19 IgA samples analyzed that had been designated as 'high titer' for Spike-specific IgA exhibited significant Spike-directed neutralization capacity, wherein IgA binding endpoint titers and neutralization IC50 values were found to be significantly correlated. Of the 3 samples examined for neutralization capacity that exhibited positive but not high titer Spike-specific IgA, 2 were non-neutralizing. It should be noted that these were all

samples obtained 4–6 weeks after infection, and future samples may exhibit neutralization as the Ab response matures. These data extend the recent analyses of SARS-CoV-2 neutralization using diluted whole milk [14, 19].

Critically, our IgA durability analysis using 28 paired samples obtained 4–6 weeks and 4–10 months after infection revealed that for all donors, Spike-specific IgA titers persisted for as long as 10 months, a finding that is highly relevant for protection of the breastfeeding infant over the course of lactation, and also pertinent to the size of a potential donor pool for collection of milk from COVID-19-recovered donors for therapeutic use of extracted milk IgA. Notably, even after 7–10 months, only 5 of 14 samples exhibited >10% decrease in specific IgA endpoint titers, while 8 of 14 samples actually exhibited an increase in specific IgA titer. These highly durable or even increased titers may be reflective of long-lived plasma cells in the GALT and/or mammary gland, as well as continued antigen stimulation in these compartments, possibly by other human coronaviruses, or repeated exposures to SARS-CoV-2.

Given the present lack of knowledge concerning the potency, function, durability, and variation of the human milk immune response not only to SARS-CoV-2 infection, but across this understudied field in general, the present data contributes greatly to filling immense knowledge gaps and furthers our work towards *in vivo* efficacy testing of extracted milk Ab in the COVID-19 pandemic context and beyond.

## Limitations of study

One limitation to this study is that all samples were obtained from participants living in the USA, and it should be noted that those in unique geographic areas may exhibit differential immune responses. Notably, 67% of COVID-19-recovered participants reported their race/ethnicity as white or Caucasian, and therefore this sample set is not diverse enough to be considered representative of the USA as a whole. More work needs to be done to obtain sufficient samples from non-white participants. Additionally, the longitudinal and functional components of these data were conducted on small number of samples, and further study will produce a more complete and accurate analysis. Neutralization and other functional analyses for all Ab classes also must be studied in follow-up samples. As well, this study does not demonstrate that the measured milk Ab response is protective for breastfed babies.

## Supporting information

**S1 Fig. No correlation was found between Spike-specific milk IgG and IgA or IgG and secretory antibody titers.** (A) Secretory Ab versus IgG. (B) IgA versus IgG. Endpoint titers were used in 2-tailed Spearman correlation tests. SC: secretory component.
(PDF)

**S1 Data.**
(XLSX)

## Acknowledgments

As always, we are indebted to the milk donors who make this work possible. Spike protein was generously gifted from the Krammer lab.

## Author Contributions

**Conceptualization:** Jennifer Hahn-Holbrook, Susan Zolla-Pazner, Rebecca L. Powell.

**Data curation:** Jennifer Hahn-Holbrook, Susan Zolla-Pazner, Rebecca L. Powell.

**Formal analysis:** Rebecca L. Powell.

**Funding acquisition:** Rebecca L. Powell.

**Investigation:** Alisa Fox, Benhur Lee, Rebecca L. Powell.

**Methodology:** Alisa Fox, Fatima Amanat, Kasopefoluwa Y. Oguntuyo, Benhur Lee, Rebecca L. Powell.

**Project administration:** Jessica Marino, Rebecca L. Powell.

**Supervision:** Rebecca L. Powell.

**Writing – original draft:** Rebecca L. Powell.

**Writing – review & editing:** Alisa Fox, Jessica Marino, Fatima Amanat, Kasopefoluwa Y. Oguntuyo, Jennifer Hahn-Holbrook, Benhur Lee, Susan Zolla-Pazner, Rebecca L. Powell.

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
