## [Decision Letter · Decision Letter 0]

15 Jul 2021

PONE-D-21-10237

The Spike-specific IgA in milk commonly elicited after SARS-CoV-2 infection is concurrent with a robust secretory antibody response, exhibits neutralization potency strongly correlated with IgA binding, and is highly durable over time

PLOS ONE

Dear Dr. Powell,

Thank you for submitting your manuscript to PLOS ONE. After careful consideration, we feel that it has merit but does not fully meet PLOS ONE’s publication criteria as it currently stands. Therefore, we invite you to submit a revised version of the manuscript that addresses the points raised during the review process.

It is important that you carefully answer the questions raised by both reviewers reviewers 

We look forward to receiving your revised manuscript.

Kind regards,

Xia Jin, MD, PhD

Academic Editor

PLOS ONE

Journal Requirements:

3)  Please provide additional details regarding participant consent. In the ethics statement in the Methods and online submission information, please ensure that you have specified whether consent was informed.

4) PLOS ONE requires experimental methods to be described in enough detail to allow suitably skilled investigators to fully replicate and evaluate your study. See https://journals.plos.org/plosone/s/submission-guidelines#loc-materials-and-methods for more information.

To comply with PLOS ONE submission guidelines, in your Methods section, please provide a more detailed description of your methodology, specifically for assays measuring IgA, IgG, and secretory-type Abs. Please ensure that you describe the sources and catalog numbers (if applicable) of all kits, proteins, antibodies, equipment, etc. in the methods section of your manuscript.

5) In your Methods section, please provide additional information about the participant recruitment method and the demographic details of your participants. Please ensure you have provided sufficient details to replicate the analyses such as:

a) the recruitment date range (month and year),

b) where patients were recruited from,

c) a table of relevant demographic details,

d) a statement as to whether your sample can be considered representative of a larger population, and

e) a description of how participants were recruited.

6)  In your Methods section, please provide additional details regarding the cell lines and reporter viruses used in your study. Please include the source from which you obtained the cells or virus, the catalog numbers if applicable, whether the cell line was verified, and if so, how it was verified. For more information on PLOS ONE's guidelines for research using cell lines, see https://journals.plos.org/plosone/s/submission-guidelines#loc-cell-lines.

7) We note that you have included the phrase “data not shown” in your manuscript. Unfortunately, this does not meet our data sharing requirements. PLOS does not permit references to inaccessible data. We require that authors provide all relevant data within the paper, Supporting Information files, or in an acceptable, public repository. Please add a citation to support this phrase or upload the data that corresponds with these findings to a stable repository (such as Figshare or Dryad) and provide and URLs, DOIs, or accession numbers that may be used to access these data. Or, if the data are not a core part of the research being presented in your study, we ask that you remove the phrase that refers to these data.

Reviewers' comments:

Reviewer's Responses to Questions

**Comments to the Author**

1. Is the manuscript technically sound, and do the data support the conclusions?

Reviewer #1: Yes

Reviewer #2: Yes

2. Has the statistical analysis been performed appropriately and rigorously? 

Reviewer #1: Yes

Reviewer #2: Yes

3. Have the authors made all data underlying the findings in their manuscript fully available?

Reviewer #1: No

Reviewer #2: Yes

4. Is the manuscript presented in an intelligible fashion and written in standard English?

Reviewer #1: Yes

Reviewer #2: Yes

5. Review Comments to the Author

Reviewer #1: This manuscript reported that secretory IgA response was dominant among the human milk samples and the duration of IgA was 4-10 months. The authors also observed that IgA binding (ELISA) and neutralization capacities (spike-pseudotyped VSV) were positively correlated. These findings have already reported in previous studies. The authors have confirmed the previous studies but the new key findings that increase the knowledge in the field of SARS-CoV-2-specific antibodies in human milk are missed in this study.

-Pace RM, Williams JE, Järvinen KM, Belfort MB, Pace CD, Lackey KA, Gogel AC, Nguyen-Contant P, Kanagaiah P, Fitzgerald T, Ferri R. Characterization of SARS-CoV-2 RNA, antibodies, and neutralizing capacity in milk produced by women with COVID-19. Mbio. 2021 Feb 23;12(1).

-Demers-Mathieu V, DaPra C, Mathijssen GB, Medo E. Previous viral symptoms and individual mothers influenced the leveled duration of human milk antibodies cross-reactive to S1 and S2 subunits from SARS-CoV-2, HCoV-229E, and HCoV-OC43. Journal of Perinatology. 2021 Mar 1:1-9.

Concern about this manuscript:

Abstract

1. The title and abstract are extremely long (PLOS One: 250 characters for title). Authors should summarize the most important findings for the abstracts (PLOS One: 300 word max). In most of peer-reviewed Journals, references are not present in the abstract.

2. “antibodies bearing secretory component” seem a strange and not appropriate term to describe secretory antibodies (line 38).

Background

1. Infants younger than 6 months of postnatal age, cannot be vaccinated due to their immature immune system. It is why influenza and pertussis vaccines are given after 6 months.

2. Mistake “ia” should be “is” (line 74)

Study participants

1. The criteria of inclusion and exclusion for the donors are missing in the methods.

2. How the authors determined the sample size?

3. Do all donors had a clinical /instrumental diagnosis of COVID-19 infection (COVID-19 PCR test)? This information should be presented in the method.

4. Donors and Control groups are not adequately described. All the clinical characteristics of the participants are missing. The demographic description is critical as these maternal factors influence the breast milk antibody titers and neutralizing activity between mothers.

Analytical Methods

1. Control negative with only media and control negative with milk with low/absent SIgA activity (heat-treated human milk) were performed as controls in the experiment? These controls are critical as human milk contains other antimicrobial components that could reduce the viral infectivity.

2. How the sample size calculation was done to obtain good power?

Discussion

1. Authors should explain why they selected 4-6 weeks and 4-10 months post-infection to evaluate antibody titers and neutralizing activity.

2. Limitations of this study is missing and should be included.

Figure.

1. Asterisk to show difference between groups are missing on all Figures.

2. Authors should also add the statistical analysis in the figure legend and the sample size of each group.

Table

Add a table with the demographic description of the participant

Reviewer #2: Summary

In the submitted manuscript, Fox et al report on the milk antibody response specific to the SARS-CoV-2 spike protein following maternal infection. This report greatly expands on the group’s previous report by recruiting a larger cohort of individuals (n=75 participants) and collection of milk samples at 4-6 weeks and 4-10 months post infection. This study provides valuable data on the longer-term durability of the milk antibody response following maternal COVID-19. Further, it also expands on other studies that have examined the ability of human milk to neutralize SARS-CoV-2 by isolating IgA from milk and demonstrating it to be a key factor of milk that effectively neutralizes SARS-CoV-2. This work is of both biological and clinical significance.

Abstract

Per journal guidelines, the abstract should not contain citations. Some of these references are never mentioned in the Introduction (e.g., refs 3-5) and I would suggest including them there.

The concluding sentence is very long and should be restated.

Introduction

Line 63 – should “previously-infected” be hyphenated?

Line 74 – “ia” – typo

Line 71 – “Notably, after two hours in the infant stomach, the total IgA concentration decreases by <50%, while IgG concentration decreases by >75%; importantly, though total SC concentration decreases by ~60%, there ia no decrease in the stomach of infants born pre-term (within the first 3 months of life) – a population highly vulnerable to infection” – This sentence is confusing as it appears to be discussing decreased in IgA, but then ends with stating that there is no decrease in preterm infants (as measured during the first 3 months of life). The reference cited seems to indicate that there is a larger decrease in IgA in preterm infants compared to term infants, but IgG and IgM appear more stable. Please clarify.

Methods

Were any of the participants/data from the group’s previous publication included in this work, if so, can you include that information here. More details should be provided on where (general region is fine) and when sample collections took place, per IRB stipulations.

Line 91 – Can you define what laboratory-confirmed infection means, e.g., PCR test?

Line 92 – Can you expand on why certain participants continued to provide samples 4-10 months after infection? Were these convenience samples?

Line 133 – Please give more information on the antibodies used, e.g., vendor and catalog number.

Line 133/138 – first use of OD and RLU, please define.

Results

Line 119 – Please provide the catalog number for the peptide M agarose beads.

Line 149 – “Skimmed acellular milk was aliquoted and frozen at -80o C until testing. Undiluted milk samples obtained 4-6 weeks post-infection from 75 COVID-19-recovered donors, and 20 pre-pandemic milk samples obtained prior to December, 2019 were screened in our IgA ELISA against recombinant trimeric SARS-CoV-2 Spike.” This is redundant with the methods and can be omitted.

Line 151 – extra comma in “December, 2019”

Lines 171/172 – The samples highlighted are missing the “b” suffix present in the figures. Please clarify.

Line 200 – Should “of the period of follow-up” be “at follow-up”?.

Line 200 – Can the authors comment on the appropriateness of performing statistics comparing mean endpoint titers by pooling samples collected from 4-10 mo post-infection? Were other statistical tests incorporating the interval between collections considered?

Line 201 – Figure 3, panel A would be improved by making it more apparent which donors saw increases, decreases, and no change in IgA over time by coloring them by these factors. Similarly, Panel B would be improved by coloring donors by the groups highlighted in lines 206-208.

Line 204 – Even if not showing the data, please include the number of samples included in the subset. Compared to earlier figures, Fig. 4 could include these data as an additional panel. I would suggest showing these results or highlighting them in panel B (if colored as mentioned above they could be further distinguished with a “#” above the bars or below each joint ID).

Line 209 – “as with the larger durability cohort” – I thought these donors were part of this cohort, but were they not? Please clarify or rephrase.

Lines 220 – Were the milk samples tested the 4-6 week samples or the 4-10 month samples, please clarify.

Discussion

Line 266 – While the milk IgG response is very likely to be less robust than the milk IgA response, this was not demonstrated in the current manuscript, and has yet to be demonstrated with SARS-CoV-2. Consider omitting this statement on the robustness of IgG or provide a citation.

A Limitations section is missing. There is a lack of participant characteristics detailed in this current manuscript. As such it is difficult to determine if these results are generalizable to all lactating women or limited to specific demographics. Another limitation might be that IgG was not isolated and tested for the ability to neutralize SARS-CoV-2.

Figures

The figures would be improved by removing the full sentence descriptions within panels and instead only listing the most pertinent information when necessary, e.g., Fig. 2, panels A and B could be “Spike-specific secretory component” and “Spike-specific IgG”, respectively.

Fig. 1, panel B – the title over the plot indicates that prepandemic controls are included here, but this differs from the legend. Please clarify.

Fig. 1, panel C – Consider ordering the milk samples along the y-axis by endpoint titer (rather than ID). This would greatly aid in easily visualizing the amounts of samples with endpoint titers crossing each threshold.

Error bars should be defined.

Minor

The title could be improved by making it more concise. It is also somewhat circular to state “The Spike-specific IgA in milk elicited after SARS-Cov-2 infection is … strongly correlated with IgA binding”; not much would be lost by omitting some of the more declaratory adjectives.

The v in “Cov” in the title of the manuscript should be capitalized.

6. PLOS authors have the option to publish the peer review history of their article (what does this mean?). If published, this will include your full peer review and any attached files.

Reviewer #1: No

Reviewer #2: No

---

## [Author Response · Author response to Decision Letter 0]

7 Oct 2021

Editor comments: 

References have been checked.

Manuscript has been formatted.

3) Please provide additional details regarding participant consent. In the ethics statement in the Methods and online submission information, please ensure that you have specified whether consent was informed.

Details have been added in methods to indicate this was informed consent.

4) PLOS ONE requires experimental methods to be described in enough detail to allow suitably skilled investigators to fully replicate and evaluate your study. See https://journals.plos.org/plosone/s/submission-guidelines#loc-materials-and-methods for more information.

To comply with PLOS ONE submission guidelines, in your Methods section, please provide a more detailed description of your methodology, specifically for assays measuring IgA, IgG, and secretory-type Abs. Please ensure that you describe the sources and catalog numbers (if applicable) of all kits, proteins, antibodies, equipment, etc. in the methods section of your manuscript.

Methodology has been more clearly described, with catalogue numbers added.

5) In your Methods section, please provide additional information about the participant recruitment method and the demographic details of your participants. Please ensure you have provided sufficient details to replicate the analyses such as:

a) the recruitment date range (month and year),

b) where patients were recruited from,

c) a table of relevant demographic details,

d) a statement as to whether your sample can be considered representative of a larger population, and

e) a description of how participants were recruited.

Information on all participants has been added as requested in Table 1. Method of recruitment and date range have been added to the methods section. We now state that given the diversity of participant ages and stages of lactation, this study sample can be considered representative of a larger population. Notably, 67% of COVID-19-recovered participants reported their race/ethnicity as white or Caucasian, and therefore this sample set is not diverse enough to be considered representative of the USA as a whole. More work needs to be done to obtain sufficient samples from non-white participants.

6) In your Methods section, please provide additional details regarding the cell lines and reporter viruses used in your study. Please include the source from which you obtained the cells or virus, the catalog numbers if applicable, whether the cell line was verified, and if so, how it was verified. For more information on PLOS ONE's guidelines for research using cell lines, see https://journals.plos.org/plosone/s/submission-guidelines#loc-cell-lines.

Cell line and reporter virus information has been added as requested in the methods section.

7) We note that you have included the phrase “data not shown” in your manuscript. Unfortunately, this does not meet our data sharing requirements. PLOS does not permit references to inaccessible data. We require that authors provide all relevant data within the paper, Supporting Information files, or in an acceptable, public repository. Please add a citation to support this phrase or upload the data that corresponds with these findings to a stable repository (such as Figshare or Dryad) and provide and URLs, DOIs, or accession numbers that may be used to access these data. Or, if the data are not a core part of the research being presented in your study, we ask that you remove the phrase that refers to these data.

References to data not shown have been replaced with supplemental data (S1 Fig) as well as addition of data values to description of Fig 3 and the figure itself.

Reviewer Comments:

Reviewer #1: This manuscript reported that secretory IgA response was dominant among the human milk samples and the duration of IgA was 4-10 months. The authors also observed that IgA binding (ELISA) and neutralization capacities (spike-pseudotyped VSV) were positively correlated. These findings have already reported in previous studies. The authors have confirmed the previous studies but the new key findings that increase the knowledge in the field of SARS-CoV-2-specific antibodies in human milk are missed in this study.

-Pace RM, Williams JE, Järvinen KM, Belfort MB, Pace CD, Lackey KA, Gogel AC, Nguyen-Contant P, Kanagaiah P, Fitzgerald T, Ferri R. Characterization of SARS-CoV-2 RNA, antibodies, and neutralizing capacity in milk produced by women with COVID-19. Mbio. 2021 Feb 23;12(1).

-Demers-Mathieu V, DaPra C, Mathijssen GB, Medo E. Previous viral symptoms and individual mothers influenced the leveled duration of human milk antibodies cross-reactive to S1 and S2 subunits from SARS-CoV-2, HCoV-229E, and HCoV-OC43. Journal of Perinatology. 2021 Mar 1:1-9.

Neither of these studies mentioned by the reviewer measured secretory antibody nor did they measure neutralization of extracted IgA from milk, only the milk itself, therefore not demonstrating the neutralization was IgA-mediated. The Pace et. al. study looked very early after infection with no long-term follow up. The Demers-Mathieu et. al. study seems to not be on any confirmed positive donors, the sample number is very small, and it only has 4 months of follow up.

Concern about this manuscript:

Abstract

1. The title and abstract are extremely long (PLOS One: 250 characters for title). Authors should summarize the most important findings for the abstracts (PLOS One: 300 word max). In most of peer-reviewed Journals, references are not present in the abstract. 

Title has been shortened and abstract is under 300 words.

2. “antibodies bearing secretory component” seem a strange and not appropriate term to describe secretory antibodies (line 38). 

This has been corrected.

Background

1. Infants younger than 6 months of postnatal age, cannot be vaccinated due to their immature immune system. It is why influenza and pertussis vaccines are given after 6 months.

Infants are immunized against various pathogens within the first 6 months, including hepatitis, diphtheria, pertussis, tetanus, and others. We have not changed the mention of possible infant vaccines. See the CDC vaccine schedule: https://www.cdc.gov/vaccines/schedules/hcp/imz/child-adolescent.html#birth-15

2. Mistake “ia” should be “is” (line 74) 

This has been corrected.

Study participants

1. The criteria of inclusion and exclusion for the donors are missing in the methods. 

This has been more clearly described in methods

2. How the authors determined the sample size?

This has been added to the methods sections: To estimate the proportion (p) of all COVID-19-recovered milk donors that would exhibit positive IgA titers against SARS-CoV-2 in their milk after infection, we determined based on the reported IgG seroconversion rate of 90% after mild SARS-CoV-2 infection (1), that the precision (d) of the 95% confidence interval (CI) for p (CI=[p+-d]), as a function of the cohort size N of 74 would allow us to estimate p with 6.79% error. In terms of the cohort size N of 20 for milk IgG and secretory Ab analyses, this would allow us to estimate p with 13.15% error.

3. Do all donors had a clinical /instrumental diagnosis of COVID-19 infection (COVID-19 PCR test)? This information should be presented in the method. 

This has been added to methods to clarify all participants had a PCR confirmed infection

4. Donors and Control groups are not adequately described. All the clinical characteristics of the participants are missing. The demographic description is critical as these maternal factors influence the breast milk antibody titers and neutralizing activity between mothers.

Now see Table 1, where this information has been added for the COVID-19 and control groups, including race/ethnicity, age, months post-partum at 1st sample, and state of residence. We also now mention in methods that all participants were either asymptomatic or experienced mild-moderate symptoms of COVID-19 that were managed at home.

Please note that in adding this information it was determined that 1 participant (COV166) was originally included in the COVID+ group in error, and this participant never tested positive. This participant’s data was therefore excluded, and the total number of COVID+ participants in this manuscript is now 74.

Analytical Methods

1. Control negative with only media and control negative with milk with low/absent SIgA activity (heat-treated human milk) were performed as controls in the experiment? These controls are critical as human milk contains other antimicrobial components that could reduce the viral infectivity. 

Unclear if this is referring to the neutralization data? These experiments were done on extracted milk IgA, not with milk which can exhibit notable ‘background’ neutralization.

2. How the sample size calculation was done to obtain good power?

See above.

Discussion

1. Authors should explain why they selected 4-6 weeks and 4-10 months post-infection to evaluate antibody titers and neutralizing activity. 

This post-infection window was selected so as to minimize any contact with participants or their samples when they might have been contagious to the research team, while still capturing the reported peak period for SARS-CoV-2 Ab responses (2). As little longitudinal mucosal Ab data in COVID-19-recovered individuals past 3 months has been reported to date, the ≥4 month time point was selected, and as many samples that were available were used. This info has been added to methods.

2. Limitations of this study is missing and should be included. 

This section has been added after the discussion.

Figure.

1. Asterisk to show difference between groups are missing on all Figures. 

This has been added in all relevant figures.

2. Authors should also add the statistical analysis in the figure legend and the sample size of each group. 

This has been added in all relevant figure legends and sample size has been added in the legends also.

Table

Add a table with the demographic description of the participant 

Table has been added (Table 1)

Reviewer #2: Summary

Abstract

Per journal guidelines, the abstract should not contain citations. Some of these references are never mentioned in the Introduction (e.g., refs 3-5) and I would suggest including them there.

Citations have been removed.

The concluding sentence is very long and should be restated. 

This sentence has been shortened.

Introduction

Line 63 – should “previously-infected” be hyphenated? 

Corrected.

Line 74 – “ia” – typo 

Corrected.

Line 71 – “Notably, after two hours in the infant stomach, the total IgA concentration decreases by <50%, while IgG concentration decreases by >75%; importantly, though total SC concentration decreases by ~60%, there ia no decrease in the stomach of infants born pre-term (within the first 3 months of life) – a population highly vulnerable to infection” – This sentence is confusing as it appears to be discussing decreased in IgA, but then ends with stating that there is no decrease in preterm infants (as measured during the first 3 months of life). The reference cited seems to indicate that there is a larger decrease in IgA in preterm infants compared to term infants, but IgG and IgM appear more stable. Please clarify. 

This sentence has been clarified with the reference to preterm infants removed as it is not particularly relevant to the present study.

Methods

Were any of the participants/data from the group’s previous publication included in this work, if so, can you include that information here. 

10 COVID-19-recovered (COV101-COV117) and 10 pre-pandemic control (NEG046-NEG059) participants included in the present study also had their Spike IgA ELISA data reported in the our pilot study publication (3). This info has been added to the methods section.

More details should be provided on where (general region is fine) and when sample collections took place, per IRB stipulations. 

See table 1 which has been added.

Line 91 – Can you define what laboratory-confirmed infection means, e.g., PCR test? 

This has been defined now as PCR test confirmation

Line 92 – Can you expand on why certain participants continued to provide samples 4-10 months after infection? Were these convenience samples? 

This has been clarified in methods. Participants were asked to if able and willing, to continue to pump and save monthly milk samples after the initial sample as part of our longitudinal analysis. If any of the 75 participants included in this study submitted longitudinal samples at least 4 months after their initial sample, those samples were also included in the present analysis.

Line 133 – Please give more information on the antibodies used, e.g., vendor and catalog number. 

This information has been added

Line 133/138 – first use of OD and RLU, please define. 

This is now defined in the text.

Results

Line 119 – Please provide the catalog number for the peptide M agarose beads. 

This information has been added

Line 149 – “Skimmed acellular milk was aliquoted and frozen at -80o C until testing. Undiluted milk samples obtained 4-6 weeks post-infection from 75 COVID-19-recovered donors, and 20 pre-pandemic milk samples obtained prior to December, 2019 were screened in our IgA ELISA against recombinant trimeric SARS-CoV-2 Spike.” This is redundant with the methods and can be omitted. 

This has been removed

Line 151 – extra comma in “December, 2019” 

Corrected

Lines 171/172 – The samples highlighted are missing the “b” suffix present in the figures. Please clarify. 

Corrected

Line 200 – Should “of the period of follow-up” be “at follow-up”?. 

Clarified to say ‘All donors exhibited persistently significant Spike-specific IgA titers at the follow-up time point’

Line 200 – Can the authors comment on the appropriateness of performing statistics comparing mean endpoint titers by pooling samples collected from 4-10 mo post-infection? Were other statistical tests incorporating the interval between collections considered? 

Samples were not pooled. These are individual values obtained at 2 discreet time points. The data from these time points were grouped to perform an appropriate paired t-test of significance. If the kinetics of the response are further examined for intervening time points, other statistical methods will be employed. 

Line 201 – Figure 3, panel A would be improved by making it more apparent which donors saw increases, decreases, and no change in IgA over time by coloring them by these factors. Similarly, Panel B would be improved by coloring donors by the groups highlighted in lines 206-208. 

The lines and bars have been color coded as suggested.

Line 204 – Even if not showing the data, please include the number of samples included in the subset. Compared to earlier figures, Fig. 4 could include these data as an additional panel. I would suggest showing these results or highlighting them in panel B (if colored as mentioned above they could be further distinguished with a “#” above the bars or below each joint ID). The number of samples in 7-10 month subset has been added. Not clear what data the reviewer is referring to to include in Fig 4?

Line 209 – “as with the larger durability cohort” – I thought these donors were part of this cohort, but were they not? Please clarify or rephrase. 

This has been clarified in the text.

Lines 220 – Were the milk samples tested the 4-6 week samples or the 4-10 month samples, please clarify. 

This has been clarified in the text to indicate the 4-6 week samples were used.

Discussion

Line 266 – While the milk IgG response is very likely to be less robust than the milk IgA response, this was not demonstrated in the current manuscript, and has yet to be demonstrated with SARS-CoV-2. Consider omitting this statement on the robustness of IgG or provide a citation. 

This statement has been clarified to indicate it is based on our IgG data described in the paper for a subset of 20 participants.

A Limitations section is missing. There is a lack of participant characteristics detailed in this current manuscript. As such it is difficult to determine if these results are generalizable to all lactating women or limited to specific demographics. Another limitation might be that IgG was not isolated and tested for the ability to neutralize SARS-CoV-2. 

Section was added as was participant data.

Figures

The figures would be improved by removing the full sentence descriptions within panels and instead only listing the most pertinent information when necessary, e.g., Fig. 2, panels A and B could be “Spike-specific secretory component” and “Spike-specific IgG”, respectively. 

Titles have been shortened for all figures as recommended.

Fig. 1, panel B – the title over the plot indicates that prepandemic controls are included here, but this differs from the legend. Please clarify. 

This was an error, it has been removed.

Fig. 1, panel C – Consider ordering the milk samples along the y-axis by endpoint titer (rather than ID). This would greatly aid in easily visualizing the amounts of samples with endpoint titers crossing each threshold. 

We have kept the sample ID order as this makes it easier to locate samples and compared to data in other figures

Error bars should be defined.

This has been added where missing in the legends.

Minor

The title could be improved by making it more concise. It is also somewhat circular to state “The Spike-specific IgA in milk elicited after SARS-Cov-2 infection is … strongly correlated with IgA binding”; not much would be lost by omitting some of the more declaratory adjectives. 

Title has been made more concise and clarified.

The v in “Cov” in the title of the manuscript should be capitalized. 

Fixed.

---

## [Decision Letter · Decision Letter 1]

2 Feb 2022

The IgA in milk induced by SARS-CoV-2 infection is comprised of mainly secretory antibody that is neutralizing and highly durable over time

PONE-D-21-10237R1

Dear Dr. Powell,

We’re pleased to inform you that your manuscript has been judged scientifically suitable for publication and will be formally accepted for publication once it meets all outstanding technical requirements.

Kind regards,

Etsuro Ito

Academic Editor

PLOS ONE

Reviewers' comments:

Reviewer's Responses to Questions

**Comments to the Author**

1. If the authors have adequately addressed your comments raised in a previous round of review and you feel that this manuscript is now acceptable for publication, you may indicate that here to bypass the “Comments to the Author” section, enter your conflict of interest statement in the “Confidential to Editor” section, and submit your "Accept" recommendation.

Reviewer #2: All comments have been addressed

2. Is the manuscript technically sound, and do the data support the conclusions?

Reviewer #2: Yes

3. Has the statistical analysis been performed appropriately and rigorously? 

Reviewer #2: Yes

4. Have the authors made all data underlying the findings in their manuscript fully available?

Reviewer #2: Yes

5. Is the manuscript presented in an intelligible fashion and written in standard English?

Reviewer #2: Yes

6. Review Comments to the Author

Reviewer #2: The authors have addressed all of my comments. The current version of the manuscript reads well and I have no further comments.

7. PLOS authors have the option to publish the peer review history of their article (what does this mean?). If published, this will include your full peer review and any attached files.

Reviewer #2: No

---

## [Editor Report · Acceptance letter]

14 Feb 2022

PONE-D-21-10237R1 

The IgA in milk induced by SARS-CoV-2 infection is comprised of mainly secretory antibody that is neutralizing and highly durable over time 

Dear Dr. Powell:

I'm pleased to inform you that your manuscript has been deemed suitable for publication in PLOS ONE. Congratulations! Your manuscript is now with our production department. 

Kind regards, 

on behalf of

Prof. Etsuro Ito 

Academic Editor

PLOS ONE